# Deetect: A Deep Learning-Based Image Analysis Tool for Quantification of Adherent Cell Populations on Oxygenator Membranes after Extracorporeal Membrane Oxygenation Therapy

**DOI:** 10.3390/biom12121810

**Published:** 2022-12-03

**Authors:** Felix Hoeren, Zeliha Görmez, Manfred Richter, Kerstin Troidl

**Affiliations:** 1Department of Pharmacology, Max Planck Institute for Heart and Lung Research, 61231 Bad Nauheim, Germany; 2Campus Kerckhoff, Justus Liebig University Giessen, 61231 Bad Nauheim, Germany; 3Department of Life Sciences and Engineering, TH Bingen, University of Applied Sciences, 55411 Bingen am Rhein, Germany; 4Department of Cardiac Surgery, Kerckhoff Heart Center, Benekestr. 2-8, 61231 Bad Nauheim, Germany; 5Department of Vascular and Endovascular Surgery, Cardiovascular Surgery Clinic, University Hospital Frankfurt and Wolfgang Goethe University Frankfurt, 60590 Frankfurt, Germany

**Keywords:** ECMO, cell adhesion, cell quantification, confocal microscopy, deep learning

## Abstract

The strong interaction of blood with the foreign surface of membrane oxygenators during ECMO therapy leads to adhesion of immune cells on the oxygenator membranes, which can be visualized in the form of image sequences using confocal laser scanning microscopy. The segmentation and quantification of these image sequences is a demanding task, but it is essential to understanding the significance of adhering cells during extracorporeal circulation. The aim of this work was to develop and test a deep learning-supported image processing tool (Deetect), suitable for the analysis of confocal image sequences of cell deposits on oxygenator membranes at certain predilection sites. Deetect was tested using confocal image sequences of stained (DAPI) blood cells that adhered to specific predilection sites (junctional warps and hollow fibers) of a phosphorylcholine-coated polymethylpentene membrane oxygenator after patient support (>24 h). Deetect comprises various functions to overcome difficulties that occur during quantification (segmentation, elimination of artifacts). To evaluate Deetects performance, images were counted and segmented manually as a reference and compared with the analysis by a traditional segmentation approach in Fiji and the newly developed tool. Deetect outperformed conventional segmentation in clustered areas. In sections where cell boundaries were difficult to distinguish visually, previously defined post-processing steps of Deetect were applied, resulting in a more objective approach for the resolution of these areas.

## 1. Introduction

Extracorporeal membrane oxygenation (ECMO) is a widely used treatment for patients with isolated or combined lung and/or heart failure. As an invasive procedure, it represents a critical therapy that has also demonstrated its pivotal role in the treatment of critically ill intensive care patients during the SARS-CoV-2 pandemic. In ECMO, venous blood is drained from the patient’s body, oxygenated in the membrane oxygenator (MO), and returned to the venous (VV-ECMO) or arterial (VA-ECMO) system with the help of a centrifugal pump.

The strong interaction of blood with the foreign surface of this artificial circuit, especially with the MO, which provides up to 1.8 m^2^ of gas exchange interface, harbors a great potential for complications [1,2]. Although new materials for gas exchange membranes and their coatings are being developed, the membrane is not inert to immune responses [3]. As a consequence of the blood–surface interaction, a systemic inflammatory response to ECMO therapy is a common issue. In this process, activation of the blood coagulation and immune systems by the foreign oxygenator surface leads to harmful processes such as excessive leucocyte cytokine release and extravasation, which can result in end-organ damage and systemic inflammatory response syndrome (SIRS) [4,5,6]. As a part of this reaction, the adhesion of leucocytes to the foreign material of the oxygenator has not fully been characterized, and its role in systemic inflammation during ECMO remains unknown [7,8,9]. Furthermore, fluorescence microscopic examination and quantification of these cell deposits has been challenging in previous studies because of prominent cell accumulation and complex spatial arrangement of cell aggregates. Due to the need for accurate cell counting in a large number of samples (batch processing) as well as the time-consuming nature of manual quantification processes, methods to solve such issues that are based on deep-learning algorithms have become state of the art [10,11].

In order to establish the methodological fundamentals for a high-resolution, spatial investigation of cell depositions on oxygenator membranes after patient support, confocal laser scanning microscopy was used, and a deep learning-based quantification tool (Deetect) was developed. The overall aim was to directly quantify densely adhering cells, which could not be achieved so far due to difficulties in microscopy and quantification methods.

The reliability of Deetect’s segmentation and postprocessing was tested in this study by comparing the quantification results of confocal substacks (2-Z-stacking) of defined predilection sites on ECMO membranes used with a conventional segmentation approach in Fiji and a manual reference count. In an exemplary application, Deetect was used to determine cell counts at the predilection sites (“hollow fibers and warps”) of an oxygenator using confocal image sequences.

## 2. Materials and Methods

### 2.1. Experimental Dataset

#### Preparation of the Biological Samples

A phosphorylcholine-coated polymethylpentene (PMP) MO (PLS-I Getinge—Göteborg, Sweden) was examined for cell colonization after long-term patient support (>24 h) using fluorescence nuclear staining of adhering cells via 4’,6-diamidino-2-phenylindole (DAPI) and confocal laser scanning microscopy (SP5/40X, Leica—Wetzlar, Germany).

After explanting, the oxygenator was rinsed and fixed using saline, formaldehyde, and saccharose solutions according to a published protocol [8]. After the fixation process the oxygenator was cut open and membrane samples from venous, middle, and arterial parts of the oxygenator were stored at −80 °C. For microscopy, each membrane sample was cut into 4 (1 cm × 1 cm) specimens, each with 2 limiting warps (Figure 1).

### 2.2. Confocal Laser Scanning Microscopy

For each specimen, imaging was carried out with high-resolution confocal laser scanning microscopy. First, the membranes were searched for predilection sites with increased cell adherence that have been previously defined [8,12]. Nodal points of the warps, or stitches, connecting the hollow fibers and contact points of adjacent hollow fibers were identified. Five representative sites on hollow fibers and five of the hollow fiber connecting warps were then selected to perform a confocal scan (Figure 1). The Z-scan range is shown in Table 1 and represents the entire spatial extent of the cell deposits at the examined predilection sites. The raw data of this study can be reviewed on: https://edmond.mpdl.mpg.de/dataset.xhtml?persistentId=doi:10.17617/3.2OEMSK (accessed on 23 November 2022).

During the scan, the confocal system automatically traverses the staked volume in the *Z*-axis, taking images of ten individual Z-planes at equally spaced intervals. The result is an image sequence that represents the spatial arrangement of the cells in the particular deposition site. The analysis of these image sequences can be challenging when deposits include a high number of cells (Figure 2).

## 3. Results

### 3.1. Cell Quantification Methods

In biomedical research, the segmentation and enumeration of stained cells imaged by confocal microscopy are challenging, since clusters of overlapping cells and different sizes of nuclei impede the analysis (Figure 2). Furthermore, accurate manual counting is often very time consuming. To address this issue, two automated cell quantification tools, one using Fiji and one using an artificial intelligence (AI)-based approach (Deetect), were created and compared with standardized manual counting as a reference method.

#### 3.1.1. Manual Counting

Manual assessment was carried out by an investigator experienced in microscopic image analysis using a standardized procedure. The images were visually scanned, and well-defined cell nuclei were numbered to determine the number of nucleated cells. As the cell nuclei were not always exactly in the plane of the confocal section, their appearance could differ from the classical nucleus shapes. Cell nucleus conglomerates were sometimes difficult to differentiate with the human eye. To resolve these aggregates, cell nucleus borders were color-marked and then counted (Figure 3).

#### 3.1.2. Conventional Segmentation and Counting via Fiji

In this approach, a macro was programmed in Fiji [13] to automatically quantify the cell count. It was written in ImageJ macro script language. In the first two steps, the images were automatically converted into binary images using the Li-white method [14]. It was used to set a threshold for the intensity and remove background noise and artifacts from the image to emphasize the cell nucleus structures. Since areas of increased nucleus density lose their differentiability when a threshold is used, the watershed tool was applied in the next step to draw dividing lines between touching nuclei. The final step of the macro was the counting of the cell nuclei via the analyze-particles function, which scans the image until it detects the edge of an object, and then it outlines the object boundaries and measures the area. The steps are repeated until the entire image is scanned.

The minimum area of the objects to be counted was set to 20 pixels^2^. The circularity of the objects to be detected was set to 0–1 due to the different shapes of the cell nuclei. After execution of the function, the result is displayed as an overview drawing with a representation of the object outlines and a results table (Figure 4). The output of the ImageJ particle-analyzer macro was then visually inspected and compared with that of the reference method (manual counting).

#### 3.1.3. AI-Based Cell Quantification Tool–Deetect

In highly aggregated areas of the investigated samples, a low differentiation capability of individual cell nuclei was expected using the conventional semantic segmentation applied in Fiji. Therefore, an AI-based approach using Cellpose as a central component was initiated to better resolve these areas, as it was expected to perform better than conventional segmentation [11,15]. Nevertheless, a central problem in applying these methods is that they involve many subprocesses (pre-, main-, and post-processing) that are difficult to modulate by scientists who are inexperienced in programming.

To resolve this issue, a deep learning-based processing tool (Deetect) was developed for the quantification process. It has a graphical user interface (GUI) with different user-configurable features to handle problems that occurred during quantification. An overview of the different features is given in Figure 5. The subprocesses of Deetect were visualized in the output by outlining the cell borders with different colors and can be viewed in a text file containing the measurements for each nucleus. This enables the examiner to oversee the different steps of the final result. All subprocesses were tested independently.

Deetect, including information about its implementation and the used software stack, can be downloaded at: https://github.com/zgormez/Deetect/blob/main/README.md#deetect-development (accessed on 26 November 2022).

Batch processing: This feature allows the processing of multiple image sequences in a short amount of time. After opening Deetect, the users can decide whether they want to analyze one image to test the parameters or to do a batch processing of multiple image sequences to generate results for a bigger dataset. In the latter case, they can simply select a directory, and all contained sequences will be processed in the same way.Z-projection: Merging images into a maximum projection can be convenient [16]. However, for optimal cell counting, the examiner must avoid crowded projections with blurred cell contours where individual cells can no longer be distinguished. To avoid crowded projections, partial stacking is a good solution. In Deetect, there are six options for z-projection algorithms: maximum, minimum, mean, median, standard deviation, and sum. For stacking, the user can choose between the following options:5-Z-stacking, where the first and the last 5 Z-planes are summed up into two substacks; 2-Z-stacking, where the whole stack of 10 Z-planes is divided into five substacks, each containing two consecutive Z-planes summed up; and the complete projection showing all 10 Z-planes summed up.Segmentation: To segment the nuclei, a deep learning-based segmentation method called Cellpose [11] was used. It is a state-of-the-art method for cellular segmentation that has been shown to outperform other, well-known methods [10,11,17]. Cellpose was chosen because of its strong ability to generalize, which means that it is able to segment biological pictures out of the box without being trained on the specific dataset. This is attributed to the different convolutional neural networks that are provided in the program that were pretrained on a highly variable dataset of biological images with over 70,000 segmented objects [11]. In order to produce reliable results, some of the parameters and models Cellpose uses must be adapted to the dataset being analyzed. Deetect’s GUI allows the user to choose different Cellpose models and parameters to apply the best settings for their analysis.For the dataset generated in this study, the cytoplasm model from Cellpose was used because it achieved the best segmentation (data not shown). The performance parameters of Cellpose were set to flow threshold = 0.6 and mask threshold = 0.0.Double-counting analysis: One of the artifacts of partial stacking is the double-counting problem. Some nuclei are represented in two substacks after partial stacking, and these nuclei will be detected twice during the segmentation and counting process. This situation is shown in Figure 6. Because of this problem, partial stacking projections may result in unwanted double-counting of nuclei. To address this issue, a method based on colocalization analysis for the detection and elimination of such double-counting artifacts was developed. The algorithm compares all nuclei of two sequential projections (Figure 6). If a nucleus from one projection has an intersection with a nucleus from an adjacent projection that is greater than a user-defined threshold, the algorithm marks the smallest of the two nuclei as a double-counted nucleus. This user-defined threshold is called the intersection percentage (default value is 60%). The nucleus that would have been double counted is outlined in red in the resulting output image (Figure 7).

Elimination processes: Artifacts can occur in very sensitive deep-learning algorithms. In this study, Deetect was able to detect cell nuclei whose structure could not be recognized by the human eye. These counts cannot be confirmed by the investigator and must therefore be removed from the data in a subsequent exclusion procedure. Additionally, very small structures appeared that could not be identified as a cell nucleus. In this study, it was assumed that these were artifacts caused by hemolysis during long-term ECMO support [18,19]. To avoid result distortion by these unwanted small and invisible structures, elimination rules were defined. Small, artifactual structures that should not be counted can be eliminated by defining a small nucleus area limit in the GUI. Detected nuclei that cannot be verified visually were eliminated using a visibility limit. The default setting is 16 because it represents the first distinguishable color from pure black [20].The rules of the elimination processes are applied on all detected nuclei. As a result, all problematic nuclei are outlined with blue (invisible nucleus) and yellow (small nucleus) in the output image (Figure 7).
The rule for the elimination of small nuclei: the area of a nucleus is less than the user-defined small nucleus area limit.
⚬nucleus_area < small_nucleus_area_limit (default value: 100 px).
The rule for the elimination of invisible nuclei: more than 50% of the pixels of the nucleus have an intensity value less than the visible intensity.
⚬invisible_pixel: if intensity < 16 (visibility_limit).⚬percent_of_invisible_pixel > 50.

Cell block analysis: Detecting nuclear boundaries in areas of high cell accumulation is a common problem for quantification algorithms. When strong light reflections occur in places with increased cell density, nuclei borders are hard to distinguish. This can often lead to undercounting, e.g., if the cluster is counted as one nucleus instead of the actual quantity within the cell cluster. To define this type of cell cluster, a rule-based analysis was developed. After identifying those cell clusters as “big nuclei,” the algorithm approximately calculates the number of nuclei present using the average nucleus area, which was determined to be 400 px in this project. In the resulting image (Figure 7), the big nuclei are labeled with gray outlines.
The rule for big nuclei: the area of a nucleus is bigger than the user-defined big nucleus area limit.
⚬nucleus_area > big_nucleus_area_limit.⚬>50% of the pixels have an intensity > 127.⚬approximate_cell_count = big_nucleus_area/average_nucleus_area.

Statistics and visualization: Deetect allows statistical analysis and data visualization. After analyzing the image sequences, the user can group the generated data as desired in Deetect’s statistics module and compare different groups using statistical tests. In this study, the Mann–Whitney U test was performed after the cell quantification process to compare different predilection sites for increased cell deposition with each other. It is a non-parametric test that does not require a normal distribution and is therefore integrated as the default setting. For group analysis of the predilection sites from different parts of the oxygenator, the Kruskal–Wallis test and post hoc Dunn’s test were applied.

### 3.2. Evaluation of Deetect

To compare the methods, 100 2-z-substack projections were analyzed by all three procedures and evaluated afterwards. Especially problematic regions with many cell clusters were inspected visually (Figure 8). Figure 9 shows the distributional characteristics of the total cell count provided by the different methods. The quantification results are presented in Appendix A). The results of the statistical tests are annotated in the graphs. The resulting segmented images of the conventional segmentation approach in FIJI and Deetect can be found at: https://edmond.mpdl.mpg.de/file.xhtml?fileId=199594&version=3.0 (accessed on 23 November 2022).

The conventional segmentation using Fiji offers a low resolution of the individual nuclei borders, especially in clustered areas where the cell boundaries can still be visually distinguished with the human eye (green circles/Figure 8). In those areas, Deetect shows an adequate resolution of the nuclei borders. In sections that are difficult to distinguish even with the human eye, the previously defined post-processing rules of Deetect were applied. For example, larger cell clusters were recognized by Deetect as a cell block and were estimated as a larger number of cells in the post-processing (red circle/Figure 8). The distributional characteristics of the methods (Figure 9) show similarities between the manual counting and the conventional segmentation via Fiji. The distribution of the AI-based analysis shows a higher cell count in relation to the other methods. The Mann–Whitney U test indicates that the AI result differs significantly from the others.

Cellpose models that are pre-trained neural networks are deterministic in their structure. When analyzing one and the same image again and again with the same pre-trained model and parameters, the same reproducible result is obtained. In the laboratory routine, however, no exact rotation of the samples on the slides is prescribed; therefore, the rotation is subject to chance. To evaluate the influence of the rotation of the image material on the quantification by Deetect, images rotated by 90° were compared with the corresponding non-rotated images (n = 125). There was a mean deviation of 2.99% ± 2.83% between the rotated and non-rotated images. Differences in the segmentation were observed (Figure 10). A comparison of the groups using the Mann–Whitney U test showed no relevant difference between the groups (*p* = 0.9359) (Figure 10C).

### 3.3. Applicability of Deetect for the Comparison of Predilection Sites (Warps and Hollow Fibers)

In an exemplary application, Deetect was used to estimate the number of cells on densely colonized predilection sites of the Maquet PLS-I MO (warps and hollow fibers) after patient support to show its usability. For this purpose, 55 image sequences of hollow fibers and 60 image sequences of connecting warps from the oxygenator were acquired using confocal laser scanning microscopy, each containing 10 z-planes. These sequences were then processed and enumerated by Deetect using 2-Z-stacking (5 substacks). Post-processing (double counting and elimination) removed an average of 126 ± 63 cells from the original count on hollow fibers and 265 ± 135 cells on the warp threads.

On average, 569 ± 238 cells were counted on each warp thread predilection site after eliminating possible double counts and performing the post-processing steps. Subsequently, an average of 331 ± 156 cells were found on the hollow fibers. The results of the Mann–Whitney U test indicate that the number of cells on the warps and the hollow fibers differ significantly (Figure 11). We also examined different parts of the oxygenator along the blood flow. The quantification results of this analysis after the elimination processes can be seen in Appendix A).

## 4. Discussion

### 4.1. Cell Quantification Methods

To quantify adhering cells on PMP-ECMO membranes imaged via confocal laser scanning microscopy, a large amount of spatial image data needs to be processed. To this end, a batch processing function and Z-projection choice was added to Deetect. AI-based segmentations are state of the art when it comes to cell quantification in biological images. For this reason, Cellpose with its various selectable models was integrated into Deetect to generate the best-possible segmentation result for the user. Depending on the image material, different artifacts may be present that can affect the quantification process. In this study, double-counting artifacts generated by the use of substacks were eliminated by post-processing using colocalization analysis, which quantifies the spatial relationship of objects between different images. Commercial and free image analysis software often has this feature, but for the analysis the user has to create a pipeline to be able to analyze the appearance of nuclei in different substacks of the applied projection [21]. This function is integrated into Deetect to make it easier to use without further programming. An additional elimination process was required to eliminate the quantification of very small nuclear-stained structures and other structures that were recognized by Cellpose but could not be visually verified as nuclei. On the other hand, large stained nuclear structures (aggregates) were visualized whose cell number had to be estimated by defined post-processing steps.

Many image analysis platforms such as Cellprofiler and ImageJ already offer their users access to the deep learning-based tool Cellpose due to its power and widespread use [10,22,23,24,25]. However, the modification and advanced use of this tool in a pipeline is challenging for users who are not trained in programming because appropriate methods have to be established as well as parameters for each necessary subprocess.

All the abovementioned functions were successfully incorporated in Deetect. With its customizable pre-, main-, and post-processing, user-friendly GUI, and the pre-validated Cellpose method, Deetect provides an easily applicable deep learning-based solution for the analysis of confocal image sequences of cells adhering to PMP-ECMO membranes.

### 4.2. Comparison of the Methods

The results show that the tool developed in the current study, Deetect, is superior to the traditional segmentation approach in Fiji. In the visual inspection, Deetect showed better segmentation results, especially in the areas with nuclei that were still visually distinguishable. The statistical analysis of the methods showed significant differences between the cell number counted with Deetect and that determined manually or by Fiji. Although manual processing is a reference method in image processing, it is insufficient in cases where dense cell clusters are present (Figure 3C). These cell blocks are often undercounted in the visual assessment and can be a source of subjective bias (interobserver variability). Deetect improved the cell number estimation by dividing the area of the cell block by the average nucleus area in the image material and thus resolved these areas more objectively. The differences in the distributions could be explained by the fact that Deetect estimates a higher cell count in areas that are difficult to distinguish visually and tends to over-segment in general.

Rotation of the images generated small, non-significant variations in the cell count. Therefore, it can be assumed that a random rotation of images during microscopy will not have a strong influence on the results of cell quantification. The deviations are possibly due to the different starting point of algorithms that are used by Cellpose to predict horizontal and vertical gradients [11]. To improve the quality of Deetect and test it further, it needs to be applied on larger data sets.

### 4.3. Imaging and Applicability of Deetect for the Comparison of Predilection Sites

The average height of the cell deposits (50.74 ± 26.45 µm) on hollow fiber predilection sites was similar to that reported in previous studies (30–45 µm) [9]. The predilection sites investigated in the present study have already been identified, and, in the case of the hollow fibers, have been linked to increased deposition of platelets and the von Willebrand factor [9,26]. However, spatial visualization and direct quantification of adherent cells in those areas have not been reported until now. The combination of confocal laser scanning microscopy with Deetect, used in this study, proved to be a suitable method for imaging and quantifying these problematic regions. Compared with the immunofluorescence microscopy used to date, this technique is more advantageous as these predilection sites with higher cell densities can also be evaluated [8]. The predilection sites were selected on the basis of their morphology and representative cell density, which was subjectively assessed by the investigator. The introduction of an objective parameter for the selection of these sites is of outstanding importance, especially for large standardized clinical trials in which the influence of cell colonization on patient outcome is investigated.

The exemplary application of Deetect showed a good usability in these problematic areas. The quantification results indicated significant differences between the amount of cell deposition on predilection sites on hollow fibers and warps (*p* < 0.0001) in the investigated oxygenator. These findings are not representative due to the small sample size and need to be verified in prospective studies. One cause of the observed differences could be the polyfile nature of the hollow fiber-connecting warp. This could physically filter cells like a sieve and cause greater cell accumulation. In addition, the warp threads are made of polyethylenterephthalate (PET), whereas the membrane is made of polymethylpentene (PMP). However, studies from the field of cardiac surgery show good hemocompatibility of surface-modified PET [27,28,29]. In addition, the entire oxygenator is internally coated with phosphorylcholine, which improves hemocompatibility. In long-term use, however, direct interaction of immune cells with uncoated PET would be possible due to washout phenomena [3,30]. Whether the deposition of cells is affected by different properties of PET and PMP needs to be clarified. The significance of the deposited cells in connection with proinflammatory or protective processes during extracorporeal circulation is also still unclear. Deetect provides the basis for future identification and quantification of these cells in combination with immunofluorescence staining. Deetect was developed and tested on a single oxygenator and can now be used for further, more extensive studies of adherent cells on ECMO membranes.

## 5. Conclusions

In this study, the development and properties of Deetect, a deep learning-based tool for the analysis of confocal image sequences of cells adhering to PMP-ECMO membranes, were described. No data-based statement can be made about Deetect’s applicability to datasets derived from other experiments. Nonetheless, especially due to the pre-validated generalizing Cellpose algorithm, a different use in experimental approaches that also investigate preparations with large Z-volumes is conceivable. It unites all processes of the analysis from segmentation to statistical testing and is an easy way to provide the power of neural networks to scientists who are untrained in programming. The quality of Deetect has yet to be tested on larger data sets.

This study also shows that in combination with confocal laser scanning microscopy, a direct quantification of cells can be achieved with Deetect even in densely populated areas, which represents a step forward compared to the indirect quantification of adherent cells on PMP membrane preparations after ECMO therapy. In future work, cell type-specific immunostaining can be applied to characterize adherent cells in different predilection sites for increased cell deposition. The next goal after cell type characterization will be to define the role of different adhering cell types during ECMO therapy.

In the future, Deetect will be extended with various AI-based segmentation tools such as StarDist so that our users can benefit from them.

## Figures and Tables

**Figure 1 biomolecules-12-01810-f001:**
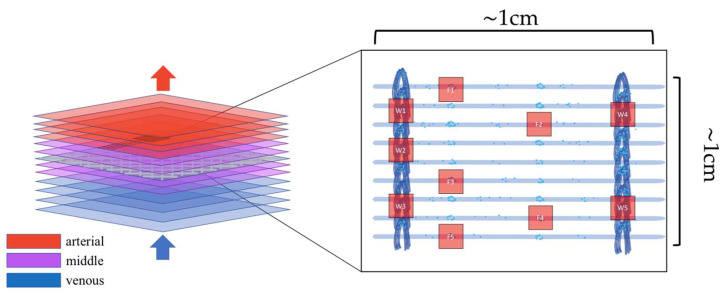
The figure shows a schematic representation of the hollow fiber mat layering within the Maquet PLS-I oxygenator (**left**). The layers are assigned to the venous, middle, and arterial sides. An examined area from the middle part of the oxygenator has been enlarged as an example (**right**). Two arrows (blue and red) indicate the direction of blood flow through the membrane stack. It shows a prepared hollow fiber mat with cell nuclei (blue dots). The entire hollow fiber mat was scanned by epifluorescence microscopy at 40× magnification, and five representative spots with increased cell deposition on the hollow fibers and the warps connecting the hollow fibers were set for confocal Z-scans (red squares).

**Figure 2 biomolecules-12-01810-f002:**
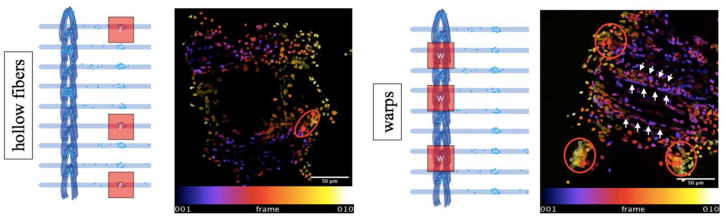
Representation of the different morphologies of cell deposits in the area of the hollow fiber and junctions of cross-linking warp threads on ECMO membranes. The representation is in temporal color code, which translates into spatial resolution. Cells from lower levels are marked in bluish hues and cells from higher levels in yellow. Z-range fiber (**left panel**) 56, 91 µm, distances between z-planes: ~5, 6 µm. Z-range warp (**right panel**): 42, 48 µm, distances between z-planes: ~4, 2 µm. F = predilection sites on fibers. W = predilection sites on warp stitches. Deposits on the warps of the oxygenator had a spatially superimposed shape with cell clusters (red arrows in red circles, right panel) and strand-like aggregates (white arrows, **right panel**). Deposits on the hollow fibers also showed a spatial overlay and close spatial proximity (red arrows in red circle, **left panel**).

**Figure 3 biomolecules-12-01810-f003:**
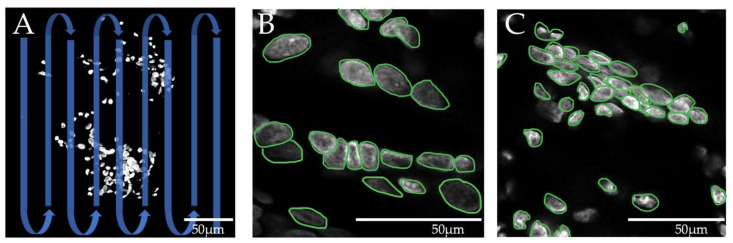
Representation of the manual counting of cell nuclei. Nuclei borders are marked with green outlines. (**A**) Meandering scanning of the confocal image. (**B**) Area with contact points of some nuclei (good delimit ability). (**C**) Cell nuclei aggregate with increasingly difficult differentiability of individual nuclei boundaries.

**Figure 4 biomolecules-12-01810-f004:**
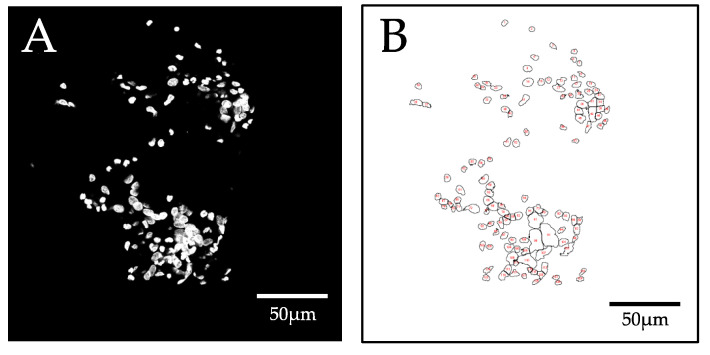
Representation of the segmentation result of the conventional segmentation approach using Fiji. (**A**) original image of a hollow fiber predilection site (**B**) segmented nuclei are outlined and annotated with a red number.

**Figure 5 biomolecules-12-01810-f005:**
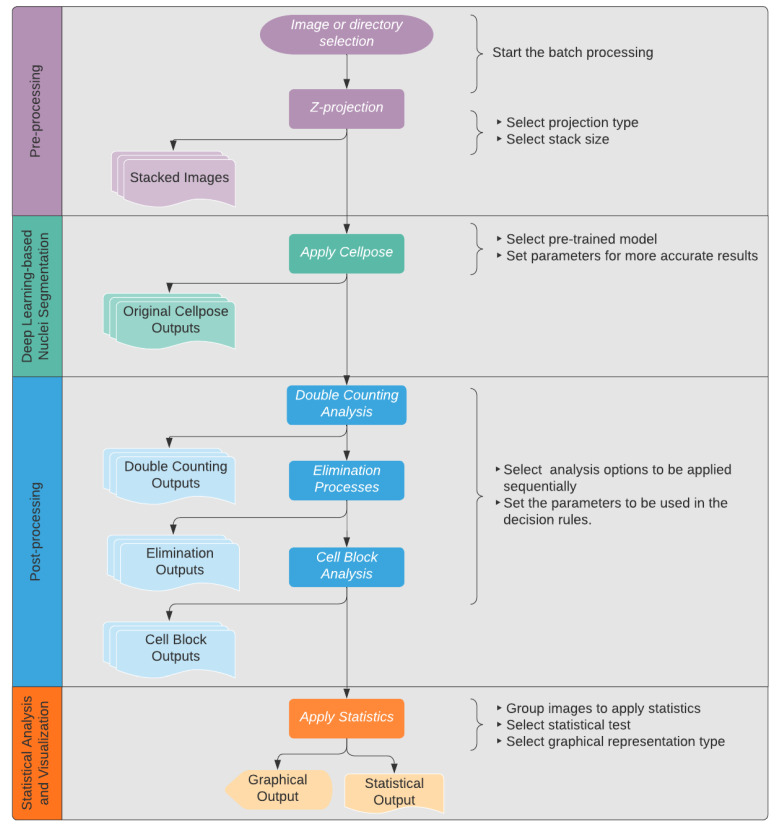
Overall flowchart representing the image analysis pipeline implemented in Deetect. The first section (pre-processing) includes batch processing and Z-projection. The outputs of this section are partially stacked images used directly as input for Cellpose. The second section is the main segmentation process performed via Cellpose. The outputs of Cellpose are a mask list that contains all detected cell nuclei and images with an outlining of the detected cell nuclei. The third section (post-processing) contains different subprocesses to achieve more accurate results. The outputs contain a color-coded graphical output and a section for statistical analysis.

**Figure 6 biomolecules-12-01810-f006:**
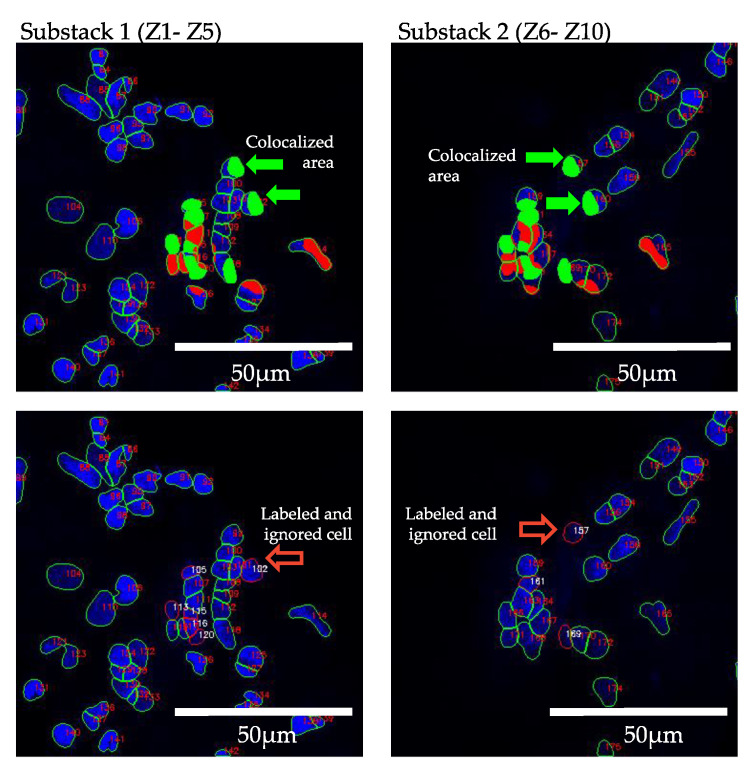
Representation of double-counting analysis. Green and red show shared areas of colocalized nuclei in two sequential projections (substack 1 contains planes 1–5, substack 2 contains planes 6–10). Red indicates that the percent of the shared area is smaller than the user-defined intersection percent (default value is 60%). Green indicates that the percent of the shared area is larger than the threshold. The nuclei outlined with red are labeled as double-counted cells and ignored in the next analysis.

**Figure 7 biomolecules-12-01810-f007:**
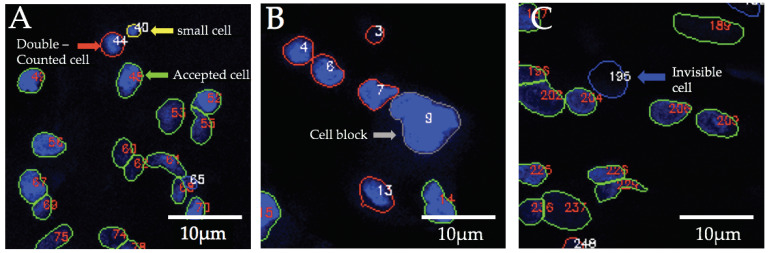
Images with cell nucleus borders outlined in different colors. The green outline indicates accepted cell nuclei that passed through all post-processing steps. The red outline represents double-counted cells concerning the previous or next projection; yellow represents a small nucleus (**A**). Blue depicts invisible cell nuclei and gray represents a cell nucleus block (**B**,**C**). Small, double-counted, and invisible cells are removed from the mask list of the related image and are not considered in the next step.

**Figure 8 biomolecules-12-01810-f008:**
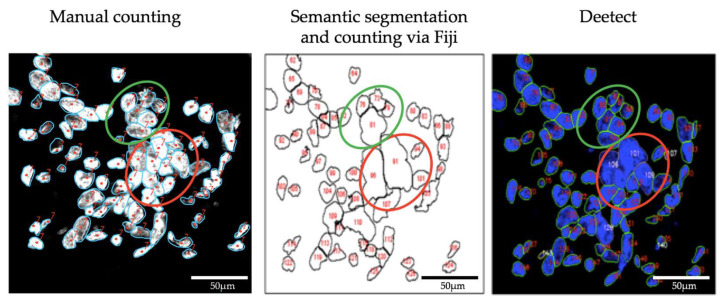
Comparison of application of the three methods to a sample region on a 2-z-substack. Regions with clustered nuclei that can still be visually distinguished (green circle, left panel) are not completely resolved by the semantic segmentation approach with Fiji (green circle, center panel) but are resolved by Deetect (green circle, right panel). In areas where it is difficult to distinguish nuclei borders visually (red circles) Deetect senses cell conglomerates (gray outlining, right panel) and estimates the number of cells present through the post-processing rules.

**Figure 9 biomolecules-12-01810-f009:**
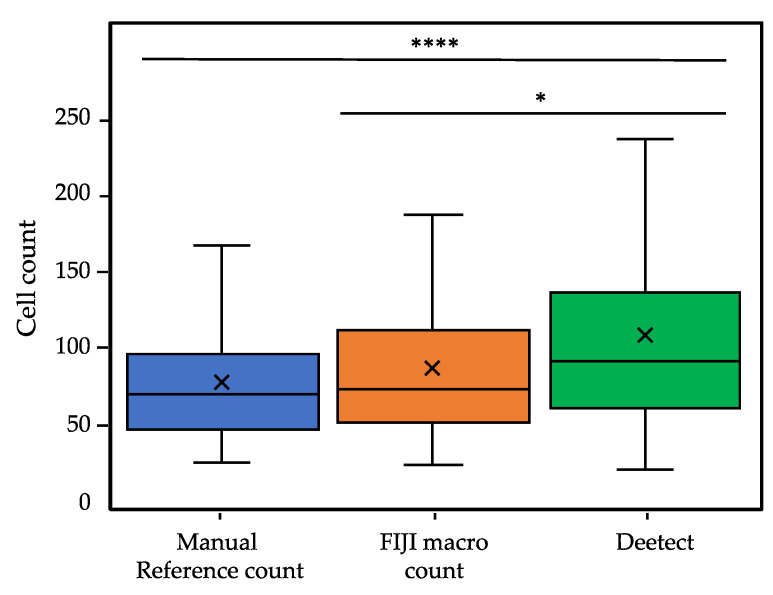
Comparison of the total number of cells detected by the 3 methods in a dataset of 100 2-Z substacks. Middle line = median; X = mean; box = interquartile range (IQR); upper/lower whisker (UW/LW) ≈1.5xIQR. The result of the Man–Whitney U test is shown using the following annotation: *, *p* ≤ 0.05; ****, *p* ≤ 0.0001.

**Figure 10 biomolecules-12-01810-f010:**
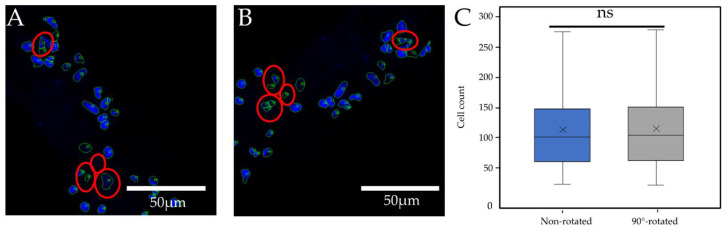
Comparison of the segmentation of a non-rotated in a 2-z-substack. (**A**) and a 90° rotated image (**B**). Differing segmentation is marked with red circles. There was no difference in the distribution of the two groups (**C**). Middle line = median; X = mean; box = interquartile range (IQR); upper/lower whisker (UW/LW) ≈1.5xIQR. The result of the Mann-Whitney-U test is shown using the following annotation: ns, *p* ≥ 0.05.

**Figure 11 biomolecules-12-01810-f011:**
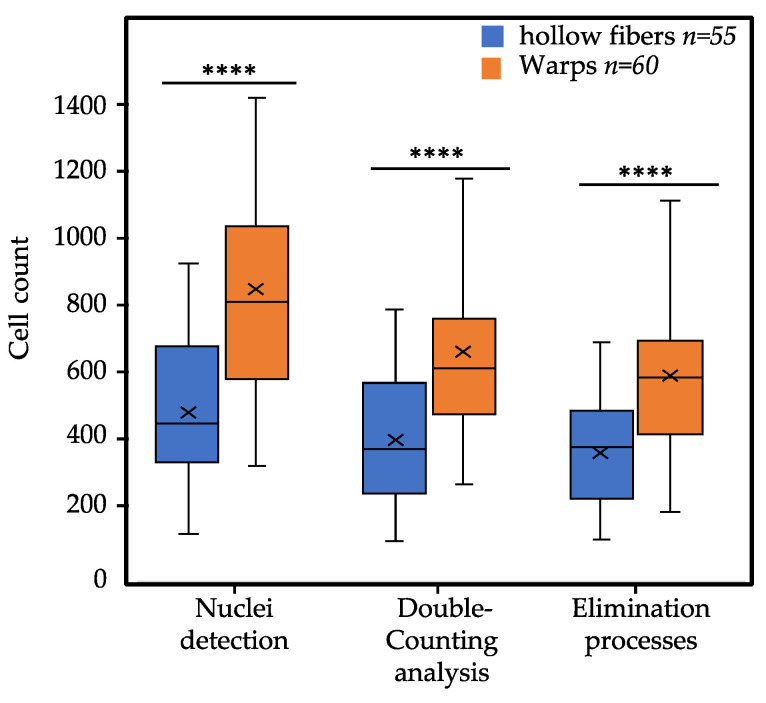
Boxplots showing the distribution of the total number of cells per predilection site after each post-processing step in Deetect. Middle line = median; X = mean; box = interquartile range (IQR); upper/lower whisker (UW/LW) ≈1.5xIQR. The result of the Mann–Whitney U test is shown using the following annotation: ****, *p* ≤ 0.0001.

**Table 1 biomolecules-12-01810-t001:** Z-scan range of cell adhesions at different predilection sites in the investigated MO.

Predilection Site	Z-Scan Range (m ± SD)
Warps	52.27 ± 25.51 µm (n = 60)
Hollow fibers	50.74 ± 26.45 µm (n = 55)

m = mean, SD = standard deviation.

## Data Availability

Deetect, a set of analyzed data and the ImageJ Fiji macro are available at: https://github.com/zgormez/Deetect (accessed on 26 November 2022) and https://edmond.mpdl.mpg.de/dataset.xhtml?persistentId=doi:10.17617/3.2OEMSK (accessed on 23 November 2022).

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
