# Peer review of "Deetect: A Deep Learning-Based Image Analysis Tool for Quantification of Adherent Cell Populations on Oxygenator Membranes after Extracorporeal Membrane Oxygenation Therapy"

_biomolecules, 2022, doi:10.3390/biom12121810_

Round 1
Reviewer 1 Report
Dear authors,
In my humble opinion, your manuscript describes an adaption of the cellpose deep-learning based instance segmentation method to streamline analysis of cell detection and counting in a simple 3D volume. You applied this method to a number of sample image stack that were acquired from samples of ECMO membranes after more than 24h of ECMO treatment of some patients. The manuscript is well written and the objectives of the study were outlined clearly. The software was easy to install and seems to run just fine on data provided with the manuscript. While I highly appreciate the work you have put in the development of a new software solution, I do have some major and some more minor points:
Major:
1. At times the main objectives/messages of the manuscript are not really clear, and I do not always see a great problem-solution fit. The manuscript would greatly benefit from you chosing a main topic: either a new software tool for 3D images analysis of ECMO membrane bound cells or characterization of the phenomena on ECMO membranes or benchmarking of 3D segmentation algorithms. Sharpening the focus of the manuscript would greatly benefit its credibility and applicability to new problems.
2. Any conclusions on the usefulness of the software outside quantifying cells on 3d imaged ECMO membranes is not supported by the presented data. Either the authors provide more data or refine their conclusions and discussion.
Minor:
1. The material methods section should contain much more detail on the software stack, its implementation, versions used and operating systems tested, just to name some examples. After all, this section should provide readers with all the information needed to recreate all methods being used.
2. A main benchmark metric used throughout the manuscript is the count of cells by different methods. Here, you assume that higher is better. I am not convinced that this number is the appropriate metric. In the state-of-the-art manual, expert counts usually serve as the gold standard to control automated deep-learning software.
3. A feature table that compares Deetect with other, similar methods would benefit the manuscript.
3. It remains up to debate whether the differences in cell occupation of different areas of the ECMO membrane provide meaningful insights. You should decide what is the focus of the manuscript. Providing a new software or understanding the biomedical impact of cell colonization of an ECMO membrane while treatment. Thus, I am not sure if figure 10 and 11 really add to the manuscript. Also, the use of statistical tests in the figures makes it hard to interpret them and understand what you want to convey by the figures.
4. I did miss the possibility to try and use different algorithms for Z-projection. As the sum is a rather unlikely always successful method, I would have liked to try different approaches like the mean, median, max-entropy, or maximum projection for cell segmentation.
5. Since programming the first version of deetect (presumably) there is a new version of cellpose (cellpose v2.1) which support full stack 3D segmentation avoiding the need to implement your own double counting avoidance algorithm as stitch thresholds are supported within cellpose. And the new version also support simply user defined creation of adapted models. It might be worthwhile trying to implement the new version and harnessing its benefits.
6. What explains (biologically) the large variance in cell occupancy of different ECMO samples or sample areas? This might be out of scope for the manuscript, yet it highlights that the choice of representative sites for imaging and analysis is crucial and special care has to be taken. How were those sites chosen then?
7. The figures seem to have been messed up during article processing:
F1: Everything, red, cannot be found in the image. Is the inlet view a top-down or side perspective on the membrane stack? How were the representative spots chosen?
F2: What is the distance between the different Z-planes? What is the meaning of the letters and numbers in the schematic? It seems that cells from different z-planes hardly ever overlay in xy-plane. Is this expected for this kind of analysis?
F3: There is a stark mismatch between the outlines and the images. Something went wrong here. Also, how were these outlines (manual) achieved? Why are images in different zoom levels. And aren't the nuclei weirdly small for lymphocytes (<5µm)? Is the projection counted or each z-plane?
F6: All highlighted cells are not visible. This is weird! Were they not originally in the image? Please carefully check the image overlays and also present the most raw data that you can in a supplemental figure.
F7: see F6
F9: Is this just one z-plane or a projection? What would be the impact on the full workflow presented using 3D imagery?
Reviewer 2 Report
Recommendation: Major
In this review, the authors introduce Deetect: a deep learning-based image analysis tool for quantification of adherent cell populations on oxygenator membranes after extracorporeal membrane oxy-genation therapy , and The reliability of Deetect was tested in this study by comparing the quantification results of confocal scans of defined predilection sites on used ECMO mem-branes with a conventional segmentation approach in Fiji and a manual reference count. This review should be greatly improved before considering its publication. Some specific comments should be pointed out.
1. The authors should give the pictures of conventional segmentation and counting via Fiji at an appropriate position in the paper, so that readers can more clearly understand the principle .
2. In the paper,do fluorescent dyes only do DAPI? Why not try using membrane dyes?
3. In the paper, Z-axis multi-layer data was used in the reconstruction. However, compared data such as manual counting and fiji did not use Z-axis data, so this experiment could not show the advantage of this technique.
4. In the paper, a cell counting method based on 3D reconstruction should be used as the control group.
5.In the paper, the authors need to provide tests on a larger dataset to verify the quality of deetdetect.
6.The authors need to more focus on the research development of recent 5 years. In the manuscript, about one thirds of references are over 5 years. I suggest the authors to adjust these references.
Round 2
Reviewer 1 Report
Dear Authors,
As far as my comments are concerned, I believe you addressed them appropriately. It is especially important that users understand which problem it is that your software tool solves for them.
Please take care when you examine the proofs of the paper before publication. Many of the Microscopy image containing figures still seem messed up. I am certain that this is neither your fault nor intentional wrong doing.
All The Best
Reviewer 2 Report
The authors have already addressed all my concerned